# Language interpretation and translation in emergency care: A scoping review protocol

Henry Li[1,2]*, Samina Ali[1,2,3], Lisa Hartling[3,4], Liz Dennett[5], Elena Lopatina[6,7], Kayathiri Ganeshamoorthy[8], Jaspreet Khangura[1]

1 Department of Emergency Medicine, Faculty of Medicine and Dentistry, University of Alberta, Edmonton, Alberta, Canada, 2 Department of Pediatrics, Faculty of Medicine and Dentistry, University of Alberta, Edmonton, Alberta, Canada, 3 Women and Children's Health Research Institute, Faculty of Medicine and Dentistry, University of Alberta, Edmonton, Alberta, Canada, 4 Alberta Research Centre for Health Evidence, Department of Pediatrics, Faculty of Medicine and Dentistry, University of Alberta, Edmonton, Alberta, Canada, 5 Geoffrey and Robyn Sperber Health Sciences Library, University of Alberta, Edmonton, Alberta, Canada, 6 Virtual Pain Program & Alberta Pain Strategy, Alberta Health Services, Calgary, Alberta, Canada, 7 Department of Community Health Sciences, Cumming School of Medicine, University of Calgary, Calgary, Alberta, Canada, 8 Provincial Interpretation and Translation Services, Alberta Health Services, Edmonton, Alberta, Canada

* henry9@ualberta.ca

## Abstract

### Introduction

Patients with preferred languages other than English face barriers to communication and access to appropriate care in English-speaking emergency care systems, leading to poorer communication and quality of care, as well as increased rates of investigations and healthcare utilization. While professional interpretation can help bridge this gap, uptake is exceedingly poor, suggesting the need for enhanced implementation and more accessible modalities. Our study will map the existing literature on interpretation/translation in emergency care, with a focus on the breadth of modalities, barriers/facilitators to implementation, and effectiveness/implementation outcomes.

### Methods

We will conduct a scoping review based on the Joanna Briggs Institute methodology. We will search MEDLINE, Embase, PsycINFO, CINAHL, Scopus, iPortal, Native Health Database and Cochrane Library CENTRAL for articles from inception to May 2024 without any language or country restrictions. Primary research articles involving interpretation/translation between English and a non-English language during emergency healthcare encounters will be included. Screening and data extraction will be completed by two independent team members. Results will be descriptively summarized and barriers/facilitators to implementation will be mapped according to the Consolidated Framework for Implementation Research.

### Stakeholder engagement & knowledge translation

Results will be disseminated at academic conferences and published in a peer-reviewed journal. We will share our key findings via a graphical abstract and social media campaign.

**Data Availability Statement:** No datasets were generated or analysed for the current study. All relevant data from this study will be made available

upon study completion. As requested by a reviewer, we are specifying that the data from this study will be made publicly available upon the review's completion.

**Funding:** The author(s) received no specific funding for this work.

**Competing interests:** Lisa Hartling is supported by a Canada Research Chair in Knowledge Synthesis and Translation and Elena Lopatina is supported by a Health System Impact Early Career Researcher Award, both from the Canadian Institutes of Health Research. The authors have no other conflicts of interest to disclose. This does not alter our adherence to PLOS ONE policies on sharing data and materials.

Our team includes our provincial health authority interpretation services lead who brings lived experience and will inform and validate our results and help identify future areas of needed research. They will also help us identify key messages and appropriate methods for dissemination to maximize knowledge translation to patients/families, local policy/clinical practice, as well as funding agencies.

# Introduction

## Background

Emergency care systems, which include prehospital care and emergency departments (EDs), provide accessible medical care to all individuals, regardless of their background. However, patients with preferred languages other than English, particularly those labeled as limited English proficiency (LEP), often face barriers to communication and access to care in English-speaking emergency care systems. This has been found to lead to increased rates of ED investigations [1–5] and healthcare utilization [1, 2]. In addition, patients with non-English preferred languages have decreased comprehension of medical instructions and reduced comfort with postdischarge care plans [6], likely leading to higher rates of unplanned ED returns and readmission [7, 8]. They also report decreased satisfaction with care and communication in the ED [9].

Professional interpretation improves outcomes for patients with LEP compared to no translation or ad hoc, untrained interpreters such as family members or members of the healthcare team [10–12]. Despite the demonstrated effectiveness of interpreters, uptake has been poor [13–16] and we have yet to understand the key barriers and facilitators of implementing different interpretation/translation services and tools. With emerging technology and increasing availability of online tools, new modes of interpretation/translation may also be able to help overcome historical barriers and improve uptake, such as machine translation tools [17, 18] and apps [19, 20]. Indeed, a recent consensus study found language translation to be one of the most desired applications of artificial intelligence by emergency physicians [21].

We are therefore seeking to map the existing literature on interpretation/translation modalities for patients in emergency care, report what key barriers/facilitators to implementation exist, and how their effectiveness and implementation is studied.

## Research questions

Our research question is: What language interpretation/translation modalities have been studied in emergency care?

Sub-questions include:

1. What type of language interpretation/translation modalities have been studied in emergency care?

2. What are the key barriers/facilitators to implementation of each type of language interpretation/translation modality in emergency care?

3. How is the effectiveness and implementation of language interpretation/translation in emergency care assessed?

## Methods

Our scoping review will be conducted based on the Joanna Briggs Institute (JBI) methodology [22]. We have registered our protocol with Open Science Framework and will report our results following the Preferred Reporting Items for Systematic Reviews and Meta-Analyses extension for Scoping Reviews (PRISMA-ScR) checklist [23]. This protocol was developed based on Peters et al's guidance on best practices in scoping review protocol development [24] and is therefore reported using the PRISMA-P checklist (S1 Appendix). Research ethics approval is not required for this scoping review.

### Information sources & search strategy

The search strategy (example in S2 Appendix) was developed and refined by a Health Sciences librarian (LD) in consultation with content and methodology experts. Peer review of the search strategy was done by a second expert librarian using the Peer Review of Electronic Search Strategies (PRESS) checklist [25].

We searched the following electronic databases from inception until May 15, 2024: MED-LINE [Ovid], Embase [Ovid], PsycINFO [Ovid], CINAHL [EBSCO], Scopus, iPortal (https://iportal.usask.ca/), Native Health Database (https://nativehealthdatabase.net/), and Cochrane Library CENTRAL [Wiley]. The search includes subject headings and free text terms for two concepts: language translation services and emergency care. For the emergency care concept, we used the filter by Campbell and Dennett, 2024 [26]. The search was optimized for use in each database. No language restrictions were included in the search. Opinion pieces, letters, editorials, and news articles were removed if possible. Results were then imported into Covidence and de-duplication was performed.

References of all included studies will be searched for additional studies. Any relevant articles/studies encountered during our work that are not captured by the systematic search will also be considered and included if they meet the eligibility criteria, following a snowballing approach.

### Article selection

Two independent reviewers will screen titles and abstracts in Covidence to identify studies relevant to this review. The two reviewers involved in any disagreement will meet to discuss and resolve through consensus. If consensus is not achieved through this process, then a third reviewer will be engaged for resolution. Subsequently, full texts of all relevant records will be retrieved and independently reviewed for their adherence to the *a priori*-defined inclusion criteria. These criteria will be finalized with team input and pilot tested prior to implementation. Disagreements at this stage will again be resolved through consensus or by a third reviewer.

Non-English texts will be included for consideration, however, due to financial limitations we will not be able to employ professional translation. As a result, we will use the team's internal resources alongside online language translation tools to translate non-English texts for selection and data extraction. Members of our study team are familiar with over 10 different languages. We will use both Google Translate and ChatGPT to translate non-English texts and compare the resulting translations. Machine translation tools provide a practical solution, however their accuracy may vary, especially in technical or medical contexts. Where discrepancies arise, they will be verified by a study team member familiar in the language of the text to ensure translation fidelity. Where no study team members are familiar with the language, we will ask a colleague who is familiar with the language to verify. In the event this is not possible we will report this as such in the limitations of the analysis.

**Table 1. Inclusion and exclusion criteria for study selection.**

| CHARACTERISTIC | INCLUSION CRITERIA | EXCLUSION CRITERIA |
|---|---|---|
| Population | Patients of all ages (pediatric and adult) with a preferred language other than English | Persons whose preferred language is English |
| | Note: the determination of preferred language may be based on various factors and it is important to consider the diverse ways it may be identified across studies. These include, but are not limited to, the individual's primary language spoken at home or self-reported preferred language | |
| Concept | Translation/interpretation of text, audio, or spoken/signed language between English and a non-English language during an emergency patient encounter | Translation/interpretation between two non-English languages or that occurs outside of an emergency patient encounter (e.g. follow-up after discharge, translation of general resources, translation of documents unrelated to direct patient care) |
| | Note: Translation/interpretation may occur bidirectionally or unidirectionally. Studies may involve the implementation of an interpretation/translation modality, evaluation of an implemented interpretation/translation modality, or both. | |
| Context | Emergency department (General, Adult, OR Pediatric), urgent care centre, or pre-hospital setting where English is the primary working language | Physician offices, in- or outpatient facilities, phlebotomy labs, dental clinics, inpatient hospital wards, long-term care homes, standardized patients/simulation/non-clinical |
| Publication Date | Inception to current | None |
| Study Design | Descriptive, observational, quantitative, qualitative, mixed methods | Editorials, letters, opinion pieces, policy recommendations, case reports, and case series |
| Publication Status | Published in a peer-reviewed journal, abstracts, conference proceedings and grey literature | Non-peer-reviewed articles, personal blogs/websites, textbooks or book chapters |
| Language | Any | None |
| Country | Any | None |

## Inclusion criteria

Peer-reviewed articles reporting original research will be included according to the PCC (population, concept, context) inclusion criteria in Table 1.

We use the terms translation and interpretation according to Allen et al's definitions [27]. Interpretation is defined as "the rendering of live utterances in one language to live utterances in another language" which includes both spoken and signed languages. Translation is defined as "movement between a recorded form of one language into a recorded form of another language" which includes written text or audio recordings of spoken language [27].

We will include descriptive, observational, quantitative, qualitative, and mixed methods studies. Editorials, letters, opinion pieces, policy recommendations, case reports, and case series will be excluded.

## Data charting

The following data will be extracted from each article:

1. Study details (e.g., first author, year of publication, country, sample size)

2. Study design (e.g., interventional, observational)

3. Setting characteristics (e.g. hospital ED, stand-alone ED/urgent care, pre-hospital setting)

4. Characteristics of interpretation/translation modality (e.g. modality, resources/technology needed, unidirectional vs bidirectional, languages involved, relevant stage of clinical care, personnel utilizing tool)

5. Participant characteristics (e.g. inclusion/exclusion, age, gender, languages spoken)

6. Outcomes including effectiveness (e.g. accuracy) and implementation (e.g. acceptability, adoption, appropriateness, cost, feasibility, fidelity, penetration, sustainability)

7. Barriers/facilitators to implementation

The data charting form will be pilot-tested using a sample of 10 full-text articles and then revised accordingly (current version in S3 Appendix). Data charting will be completed by two independent reviewers. Disagreements will be resolved through consensus or a third reviewer.

## Data synthesis

We will summarize and report the quantity, content, and coverage of the evidence. This will include 1) a summary of the language interpretation/translation modalities studied in emergency care; 2) a list of outcomes used to evaluate language interpretation/translation in emergency care; and 3) a narrative summary of key barriers and facilitators to implementation specific to each identified interpretation/translation modality, mapped to the Consolidated Framework for Implementation Research [28]. Results of included studies such as the impact of interpretation/translation on outcomes will not be extracted or assessed. Descriptive statistics will be used where appropriate, including means and standard deviations for continuous data and proportions for categorical data.

## Stakeholder engagement & knowledge dissemination

Our team includes the lead for interpretation services of our provincial health authority, Alberta Health Services (KG), who reviewed and informed the research questions for this scoping review. They also bring lived experience of navigating language barriers in healthcare and will be engaged according to the "collaborate" goals of the International Association for Public Participation spectrum for engagement [29]. Furthermore, we will be engaging additional patient partners to help inform and validate our ScR results and help identify future areas of needed research. In addition to academic knowledge dissemination venues such as publication in a peer-reviewed journal and conference presentations, we will also develop a graphical abstract and social media campaign to help share our findings more broadly. Our included stakeholders will help us identify key messages and appropriate methods for dissemination to maximize knowledge translation to patients/families, local policy/clinical practice, and funding agencies.

## Discussion

Patients with a preferred language other than English suffer disparate outcomes when accessing emergency care in English-speaking healthcare settings, leading to lower quality of care [6, 9], higher rates of investigations [1–5], and increased healthcare utilization [1, 2]. While this gap can be at least partially closed with professional interpretation [10–12], uptake is poor [13–16] and providers often try to "get by" by using a limited second language or ad hoc interpreters [30].

There is a need to better understand the barriers and facilitators to implementation of interpretation/translation in emergency care. By doing so, we can improve optimal usage and help close the care quality gap experienced by these patients. For example, single-centre quality improvement approaches have been shown to improve uptake of professional interpretation to as high as 82% [31, 32]. Through systematically mapping the literature, we hope to summarize existing barriers and facilitators to inform both local quality improvement and generalizable implementation strategies.

Furthermore, with evolving technology, there may be new modalities that can provide similar accuracy with fewer barriers to implementation. Thus far, there have been preliminary assessments of Google translate [17] and an emerging interest in the potential of artificial intelligence [33, 34], but limited implementation and real-world evidence. By exploring and mapping interpretation/translation modalities that have been studied in emergency care, we hope to understand the landscape of available options and better identify potential emerging modalities to further explore.

## Supporting information

**S1 Appendix. PRISMA-P checklist.**
(DOCX)

**S2 Appendix. Example search strategy.**
(DOCX)

**S3 Appendix. Data extraction form.**
(DOCX)

## Acknowledgments

We would like to acknowledge Lisa Tjosvold for completing the peer review of our scoping review search strategy.

## Author Contributions

**Conceptualization:** Henry Li, Jaspreet Khangura.

**Methodology:** Henry Li, Samina Ali, Lisa Hartling, Liz Dennett, Elena Lopatina, Kayathiri Ganeshamoorthy, Jaspreet Khangura.

**Project administration:** Henry Li, Liz Dennett, Jaspreet Khangura.

**Supervision:** Henry Li, Samina Ali, Jaspreet Khangura.

**Writing – original draft:** Henry Li.

**Writing – review & editing:** Henry Li, Samina Ali, Lisa Hartling, Liz Dennett, Elena Lopatina, Kayathiri Ganeshamoorthy, Jaspreet Khangura.

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
