## [Decision Letter · Decision Letter 0]

17 Sep 2024

PONE-D-24-34743Language interpretation and translation in emergency care: a scoping review protocolPLOS ONE

Dear Dr. Li,

Thank you for submitting your manuscript to PLOS ONE. After careful consideration, we feel that it has merit but does not fully meet PLOS ONE’s publication criteria as it currently stands. Therefore, we invite you to submit a revised version of the manuscript that addresses the points raised during the review process.

We look forward to receiving your revised manuscript.

Kind regards,

Atakan Orscelik

Academic Editor

PLOS ONE

Journal Requirements:

"Lisa Hartling is supported by a Canada Research Chair in Knowledge Synthesis and Translation and Elena Lopatina is supported by a Health System Impact Early Career Researcher Award, both from the Canadian Institutes of Health Research"

Reviewers' comments:

Reviewer's Responses to Questions

**Comments to the Author**

1. Does the manuscript provide a valid rationale for the proposed study, with clearly identified and justified research questions?

Reviewer #1: Yes

Reviewer #2: Yes

Reviewer #3: Yes

2. Is the protocol technically sound and planned in a manner that will lead to a meaningful outcome and allow testing the stated hypotheses?

Reviewer #1: Partly

Reviewer #2: Yes

Reviewer #3: Yes

3. Is the methodology feasible and described in sufficient detail to allow the work to be replicable?

Reviewer #1: No

Reviewer #2: Yes

Reviewer #3: Yes

4. Have the authors described where all data underlying the findings will be made available when the study is complete?

Reviewer #1: No

Reviewer #2: Yes

Reviewer #3: Yes

5. Is the manuscript presented in an intelligible fashion and written in standard English?

Reviewer #1: Yes

Reviewer #2: Yes

Reviewer #3: Yes

6. Review Comments to the Author

You may also provide optional suggestions and comments to authors that they might find helpful in planning their study.

Reviewer #1: The manuscript has a clear and sensible justification for why the scoping review should be proposed, analyzing an important issue of language barriers in emergency care among patients with LEP. The research questions are well-defined alongside the scope of mapping for partial and full existing literature on interpretation and translation services. However, some of the methodology and feasibility require further clarification to enhance this study.

Rationale and Research Questions: The rationale is sound as this would be a potential solution to address the need for improvement in professional service uptake, underpinning healthcare efficiency linked with patient health outcomes among LEP patients. The research questions address how to identify determinants for implementation, which is a topic of interest and importance in improving the quality of care.

Rigor/Soundness of the Protocol: The protocol follows the Joanna Briggs Institute (JBI) methodology for scoping reviews, a legitimate and robust standard. The search strategy is comprehensive and appropriate for the study's focus based on multiple databases. Other areas where the protocol requires some elaboration to add rigor include:

Reviewer Discrepancies: The process of disagreement resolution between reviewers (or by consensus/third reviewer) is mentioned, but what constitutes ‘consensus’ needs more explicit detail. Additional transparency could be gained by detailing the rules for involving a third reviewer (e.g., ad hoc or predefined thresholds of no consensus).

Exploratory Features: Are any aspects of the analysis exploratory? Clearly describing exploratory analysis elements could increase transparency and prevent flexibility in interpretation.

Feasibility and Replicability of the Methodology: The paper does not elaborate enough on the feasibility of this methodology, particularly:

Non-English Studies: While non-English studies refer to online translation tools, it is unclear how these will be validated for medical context accuracy. The proposed method for non-English papers could raise concerns about reliability and practicality. Clarifying what tools will be used and how word-to-word accuracy will remain unchanged is necessary. Clear outcome measures are needed, or the results risk selective interpretation, reducing replicability.

Data Extraction and Outcome Measures: More details are needed on how data will be extracted and how the effectiveness of different translation modalities will be assessed.

Data Availability: The manuscript does not clearly state where the review data will be made available, which is essential for compliance with PLOS ONE’s data policy.

Availability of Data and Materials: The manuscript does not report primary data or mention where review-related data will be made publicly available. This is necessary to satisfy PLOS ONE's data availability policy. Specify if it will be included in a public repository or as additional files (if applicable).

Suggestions for Improvement:

Reviewer Discrepancies: Clarify how disagreements between reviewers will be settled (e.g., by a third reviewer or predefined thresholds).

Exploratory Aspects: Provide detailed information on exploratory data analysis elements and any assumptions used during data processing.

Non-English Studies: Detail the translation process for non-English studies, including the tools used and how accuracy will be ensured.

Data Availability: Clearly state where the data underlying the review will be publicly available to comply with PLOS ONE’s data policy.

Conclusions: Although the protocol addresses a relevant issue, more details on the feasibility of the methodology, especially handling non-English studies and data availability, are necessary. Addressing these concerns is crucial to ensure the study is feasible and reproducible.

Recommendation: Minor Revision.

Reviewer #2: The paper entitled “Language interpretation and translation in emergency care: a scoping review protocol” is a review paper that intends to explore the challenges of language interpretation and translation in emergency care. The authors followed the e Joanna Briggs Institute (JBI) methodology to collect the intended data. Selection and seclusion criteria were described. The paper is well written and organized. All selected papers were reported and summarized properly according to the designated criteria. I do think this paper is publishable in its current status.

Reviewer #3: 1. This research is a scoping research that can be used as a base to other research.

2. Would have loved to read more in the discussion section.

7. PLOS authors have the option to publish the peer review history of their article (what does this mean?). If published, this will include your full peer review and any attached files.

Reviewer #1: No

Reviewer #2: No

Reviewer #3: No

---

## [Author Response · Author response to Decision Letter 0]

30 Oct 2024

Our response to reviewers document is attached as a table file.

---

## [Editor Report · Decision Letter 1]

5 Nov 2024

Language interpretation and translation in emergency care: a scoping review protocol

PONE-D-24-34743R1

Dear Dr. Li,

We’re pleased to inform you that your manuscript has been judged scientifically suitable for publication and will be formally accepted for publication once it meets all outstanding technical requirements.

Kind regards,

Atakan Orscelik

Academic Editor

PLOS ONE
---

## [Editor Report · Acceptance letter]

7 Nov 2024

PONE-D-24-34743R1 

PLOS ONE

Dear Dr. Li, 

I'm pleased to inform you that your manuscript has been deemed suitable for publication in PLOS ONE. Congratulations! Your manuscript is now being handed over to our production team.

Kind regards, 

on behalf of

Dr. Atakan Orscelik 

Academic Editor

PLOS ONE